

# Fertilizer and herbicide alter nectar and pollen quality with consequences for pollinator floral choices

Laura Russo[1,2], Fabian Ruedenauer[3], Angela Gronert[4], Isabelle Van de Vreken[5], Maryse Vanderplanck[6], Denis Michez[7], Alexandra Klein[4], Sara Leonhardt[3] and Jane C. Stout[2]

[1] University of Tennessee, Knoxville, United States of America
[2] Trinity College Dublin, Dublin, Ireland
[3] Technische Universität München, Freising, Germany
[4] Chair of Nature Conservation and Landscape Ecology, Albert-Ludwigs-University Freiburg, Freiburg, Germany
[5] University of Liege, Gembloux Agro-Bio Tech, Gembloux, Belgium
[6] CEFE, Univ Montpellier, CNRS, EPHE, IRD, Montpellier, France
[7] University of Mons, Mons, Belgium

Corresponding author
Laura Russo, lrusso@utk.edu

## ABSTRACT

**Background**. Pollinating insects provide economically and ecologically valuable services, but are threatened by a variety of anthropogenic changes. The availability and quality of floral resources may be affected by anthropogenic land use. For example, flower-visiting insects in agroecosystems rely on weeds on field edges for foraging resources, but these weeds are often exposed to agrochemicals that may compromise the quality of their floral resources.

**Methods**. We conducted complementary field and greenhouse experiments to evaluate the: (1) effect of low concentrations of agrochemical exposure on nectar and pollen quality and (2) relationship between floral resource quality and insect visitation. We applied the same agrochemcial treatments (low concentrations of fertilizer, low concentrations of herbicide, a combination of both, and a control of just water) to seven plant species in the field and greenhouse. We collected data on floral visitation by insects in the field experiment for two field seasons and collected pollen and nectar from focal plants in the greenhouse to avoid interfering with insect visitation in the field.

**Results**. We found pollen amino acid concentrations were lower in plants exposed to low concentrations of herbicide, and pollen fatty acid concentrations were lower in plants exposed to low concentrations of fertilizer, while nectar amino acids were higher in plants exposed to low concentrations of either fertilizer or herbicide. Exposure to low fertilizer concentrations also increased the quantity of pollen and nectar produced per flower. The responses of plants exposed to the experimental treatments in the greenhouse helped explain insect visitation in the field study. The insect visitation rate correlated with nectar amino acids, pollen amino acids, and pollen fatty acids. An interaction between pollen protein and floral display suggested pollen amino acid concentrations drove insect preference among plant species when floral display sizes were large. We show that floral resource quality is sensitive to agrochemical exposure and that flower-visiting insects are sensitive to variation in floral resource quality.

## INTRODUCTION

Although the production of many agricultural crops depends on and/or benefits from insect pollination, many pollinating insects are negatively affected by agricultural land management. For example, the agricultural yield of pollinator-dependent crops is improved by the diversity and abundance of wild pollinators (*Klein, Steffan-Dewenter & Tscharntke, 2003*; *Garibaldi et al., 2013*; *Blaauw & Isaacs, 2014*; *Dainese et al., 2019*), while the diversity and abundance of these same pollinators are negatively affected by agricultural practices related to intensification (*Kennedy et al., 2013*; *Grab et al., 2018*). The diversity and abundance of wild bees are both negatively affected by pesticide use and loss of natural habitat in the landscape surrounding farms (*Park et al., 2015*), to the extent that agricultural intensification reduces pollination services provided by wild pollinators (*Kremen, Williams & Thorp, 2002*). Pollinator populations that provide sustainable pollination services can be supported by the proximity of natural habitat (*Carvalheiro et al., 2010*; *Garibaldi et al., 2011*; *Moreaux et al., 2022*), floral resource supplementation (*Holzschuh, Dudenhöffer & Tscharntke, 2012*; *Carvalheiro et al., 2012*; *Blaauw & Isaacs, 2014*; *Von Königslöw, Fornoff & Klein, 2022*), or reduction of pesticide use (*Carvalheiro et al., 2012*; *Park et al., 2015*). Thus, pollinator conservation, in many cases, equates to attenuating the negative impacts of agriculture for the insects on which it relies.

Research continues to elucidate the mechanisms driving pollinator health (*López-Uribe, Ricigliano & Simone-Finstrom, 2020*). For example, the diet quality of pollinators contributes significantly to their health (*Alaux et al., 2010*; *Di Pasquale et al., 2013*; *Roger et al., 2017*; *Dolezal & Toth, 2018*; *Parreño et al., 2021*), although there is substantial variation in dietary preferences among species (*Leonhardt & Blüthgen, 2012*; *Kriesell, Hilpert & Leonhardt, 2017*; *Wood et al., 2021*). Pollen nutrient quality is a complex, multivariate trait (*Lau et al., 2022*), but despite species-level variation, the macronutrient composition (proteins, lipids, and carbohydrates) of floral resources seems to be consistently important across pollinator species (*Somme et al., 2015*; *Vaudo et al., 2016b*; *Moerman et al., 2017*). Specifically, the protein and lipid concentrations (and sometimes their ratio (*Vaudo et al., 2016b*) of pollen relate not only to foraging preferences (*Ruedenauer, Spaethe & Leonhardt, 2015*; *Ruedenauer, Spaethe & Leonhardt, 2016*; *Ruedenauer et al., 2019a*; *Russo et al., 2019b*), but also reproduction and fitness in bees (*Roulston & Cane, 2002*; *Ruedenauer et al., 2020*; *Centrella et al., 2020*; *Lawson, Kennedy & Rehan, 2021*).

The quantity and quality of floral resources (here, the nutritional composition) also vary across plant families and species (*Ruedenauer et al., 2019b*; *Vaudo et al., 2020*), and can even vary within plant species according to factors affecting individual plants, such as temperature (*Hoover et al., 2012*; *Takkis et al., 2015*; *Mu et al., 2015*; *Russo et al., 2019a*)
or water stress (*Descamps et al., 2018*; *Wilson Rankin, Barney & Lozano, 2020*). Exposure to agrochemicals (chemical pesticides and fertilizers) (*Burkle & Irwin, 2010*; *Dupont, Strandberg & Damgaard, 2018*) can also result in changes in floral resource quality. Given the importance of floral resource quality to pollinator foraging behaviour and fitness, this could result in significant implications for pollinator health. Thus, changes in the nutritional landscape available for pollinating insects may play a role in the observed negative effects of agriculture on pollinator populations (*Parreño et al., 2021*).

Our goals in this study were: (1) to evaluate whether low concentrations of agrochemicals, such as those found in run-off or drift, affected the quality of floral resources provided by plants to flower visitors, and (2) to determine whether patterns of flower visitation by insects correlate with variation in pollen and/or nectar quality. To this end, we combined data from complementary experimental field (*Russo et al., 2020*) and greenhouse studies on seven plant species found in weedy field-edges across Europe. Plants in the field and greenhouse studies were exposed to the same agrochemical treatment regime. We collected flower-visiting insects from the field experiment and pollen and nectar for nutritional analyses from the greenhouse experiment. We conducted the concurrent greenhouse study for the collection of pollen and nectar to avoid disturbing visitation patterns of insects in the field study.

## MATERIALS & METHODS

### Greenhouse study design

We selected seven plant species for our experiment (*Cirsium vulgare*, (Savi) Ten. Asteraceae, *Epilobium hirsutum*, L. Onagraceae, *Filipendula ulmaria*, (L.) Maxim. Rosaceae, *Hypochaeris radicata*, L. Asteraceae, *Origanum vulgare*, L. Lamiaceae, *Phacelia tanacetifolia*, Benth. Boraginaceae, and *Plantago lanceolata*, L. Plantaginaceae). These comprised six native perennial and one non-native annual herbaceous species (*P. tanacetifolia*), selected as pollinator-attractive (*Clifford, 1962*; *Russo et al., 2022*), and likely to be found on agricultural field edges in Europe. They represent a diverse group of plant families with regard to floral resource quality (*Ruedenauer et al., 2019b*; *Vaudo et al., 2020*; *Zu et al., 2021*). Of the study species, *C. vulgare, H. radicata, F. ulmaria, E. hirsutum*, and *P. tanacetifolia* offer both pollen and nectar to flower visitors, while *P. lanceolata* offers principally pollen and *O. vulgare* offers principally nectar.

In order to conduct the nutritional analyses, we aimed to collect at least 10 mg pollen and 10 mL nectar from each species and treatment combination. To collect sufficient quantities of pollen and nectar, and to avoid interrupting natural patterns of insect visitation in the field, we conducted a concurrent greenhouse study. We collected 20 wild individuals of each perennial species in the spring of 2017 and planted them in individual pots with field soil in the greenhouse. The annual species (*P. tanacetifolia*) was planted in potting media in the greenhouse with 20 seeds (purchased from a regional seed supplier: QuickCrop Ireland[©]) to each pot.

After the plants were established, we randomly assigned five individuals of each species to each treatment (see below). Treatments were applied with a watering can holding 10 litres

**Table 1 Agrochemical treatment applications for field and greenhouse experiments.** The treatments were applied foliarly once a week for three months in 10 L water.

|  | First month | Second month | Third month | Total annual application |
|---|---|---|---|---|
| N (mg/l) | 30 | 20 | 10 | 0.6 g/m$^2$ |
| P (mg/l) | 15 | 10 | 5 | 0.3 g/m$^2$ |
| K (mg/l) | 5.5 | 3 | 1 | 0.095 g/m$^2$ |
| Glyphosate (mg/l) | 0.7 | 0.3 | 0.1 | 0.011g/m$^2$ |

of water, applied across the five individuals of each of the seven species once a week. The plants were also treated with an insecticidal/fungicidal product (SB Plant Invigorator©) once a week to control pest outbreaks. The insecticidal treatment was applied evenly across all plant species and treatments.

The four experimental treatments were designed to simulate non-target agrochemical exposure on field edges: (1) control (20 L water), (2) run-off concentration of NPK fertilizer (in 10 L water plus 10 L untreated water), (3) low concentrations of herbicide (glyphosate in 10 L water plus 10 L untreated water), or (4) a combination treatment (same low concentrations of NPK in 10 L water and glyphosate in 10 L water). The treatments were mixed with 10 litres of water used for a foliar application once a week for three months. The first four weeks of application were the highest concentration, the second four weeks lower, and the last four weeks the lowest (Table 1). These applications were based on estimates of field-edge exposure; there is commonly a high concentration spring application of chemical fertilizer and herbicide, followed by decreasing exposure later in the growing season. Concentrations were selected using published studies of fertilizer run-off (*Korsaeth & Eltun, 2000*; *Bertol et al., 2007*; *Craig & Mannix, 2009*; *Russo et al., 2020*). Because glyphosate is not mobile in the ground water, we based our highest glyphosate application on the US EPA's Maximum Contaminant Level (MCL) for safe drinking water (*United States Environmental Protection Agency, 2003*). The highest concentration we applied was less than half the maximum level detected in *Silva et al. (2019)*, or roughly 7.6% of a standard annual field application (1,440 g/ha) (*Dupont, Strandberg & Damgaard, 2018*). Outside of this treatment regime, the plants in our experiments received only water.

In the greenhouse, we collected pollen and nectar daily between the hours 0600-1000. All individuals of all the treatments within each species were sampled at the same time, with the order of collections randomized on each sample day. We collected sufficient quantities of pollen for nutritional analyses (at least 10 mg per species in each treatment) from six of the seven species (all except *O. vulgare*, which produced very little pollen in the greenhouse), and sufficient quantities of nectar for nutritional analyses (at least 10 mL per species in each treatment) from three of the seven species. *Cirsium vulgare, H. radicata, F. ulmaria*, and *P. lanceolata* either did not produce nectar or had small inflorescences from which we were not able to obtain sufficient quantities of nectar. We collected nectar and pollen from greenhouse plants to avoid interrupting normal insect foraging behaviour in the field.

We counted the inflorescences from which we collected pollen and nectar on each collection day and collected pollen and nectar from every inflorescence of every individual in each treatment-species combination. Thus, we sampled a total of 20 plants (5 in each of the four treatments) for each species on each sampling day. Pollen and nectar samples from within a treatment-species combination were pooled across individuals and sampling days to generate sufficient quantities of pollen and/or nectar for analysis. Pollen was collected from dehisced anthers using forceps directly into Eppendorf tubes and transferred immediately to a −20 °C freezer. For *F. ulmaria, E. hirsutum*, and *P. tanacetifolia* we collected whole anthers; while for *H. radicata, C. vulgare*, and *P. lanceolata*, we collected fresh pollen. Both whole anthers and fresh pollen were included in the amino acid analysis, while pollen was dried and sifted for the fatty acid analysis, separating anther material from the pollen grains. We collected nectar with microcapillary tubes and measured the filled volume before transferring them to a −20C freezer. We calculated the average amount of nectar per inflorescence. Because pollen was collected fresh and later dried for analysis, we measured the total dry weight of pollen divided by the total flowers sampled for each species and treatment at the end of the season (Table S1).

## Field study design

We conducted a field experiment to measure the effects of non-target agrochemical exposure on plant growth and pollinator visitation from 2017–2018 in Dublin, Ireland (*Russo et al., 2020*). The study consisted of four experimental treatment plots (2 × 2 m) replicated across eight sites over two years (four sites in 2017 and four different sites in 2018). The sites were located in urban Dublin and selected based on space availability in collaboration with businesses and research entities, as well as the absence of outside exposure to herbicide or fertilizer. Each plot contained the same plant community with equal densities of individuals of the same seven plant species as the greenhouse experiment (above). The same experimental treatments as described above for the greenhouse experiment, with the same concentrations of fertilizer and herbicide, were used in the field experiment in both years of the study (Table 1). Treatments were randomly assigned to plots within a site at the beginning of the season. For the purposes of this study, we were primarily interested in the pollinator visitation from the field experiment.

Once the plants in the field began to flower, we sampled insects that came in contact with the reproductive parts of the inflorescences for at least 1 s (putative pollinators). On each sample day at each site, we collected flower-visiting insects on each flowering plant species at each plot for five minutes using an insect vacuum (total of 96 sample days, 623 date-plot-samples, or 2,036 five-minute samples (approximately 170 h)). Each site was visited between 12–14 times for collections in both 2017 and 2018; the number of site collections varied due to variation in the timing of flowering between different sites. We sampled between the hours of 0700 and 1800 (84% of the samples were collected between from 1000 to 1600). The order in which we visited sites, plots within sites, and species within plots was randomized during each sampling event. We also recorded the number of inflorescences of each species during each sampling event. For each plant species, we collected data on inflorescence size by randomly selecting at least 20 inflorescences and

measuring the diameter in mm at their widest and narrowest extents. We then calculated an average inflorescence size at the species level and used this value, multiplied by the number of inflorescences open during a given sample to calculate floral display for each sampling event. Insect species that could be identified in the field (specifically *Apis mellifera*, Linneaus Apidae, *Episyrphus balteatus*, De Geer Syrphidae, *Bombus pascuorum*, Scopoli Apidae, *B. lapidarius*, Linnaeus, and *B. pratorum*, Linnaeus) were released alive at the end of the sampling event. Collected specimens were transferred to a freezer and identified at the end of the field season (*Ball & Biological Records Centre, 2011*; *Falk & Lewington, 2015*). Bee identifications were verified by Dr. Úna Fitzpatrick of the National Biodiversity Data Centre (Waterford, Ireland), while hoverfly specimens were identified by Dr. Martin Speight (Trinity College Dublin, Ireland).

## Chemical analyses

We quantified amino acids in 3–6 mg of pollen of each of six plant species and four treatments and analysed three subsamples of each pollen sample for amino acids (72 samples). Here, a subsample refers to a methodological replicate to determine variance due to the method, rather than biological replication. Subsamples were 1–2 mg samples of pollen from the homogenized vial of pollen aggregated across individuals and time for each treatment-species combination. We used high-performance liquid chromatography (HPLC) and a spectrum analyser to identify the peaks of the individual amino acids, and the area under the curve of the spectra corresponded to the quantity of individual amino acids (full description in the Supplemental Materials).

We quantified fatty acids in 5–10 mg of pollen of each of six plant species and four treatments and analysed two subsamples of each pollen sample (48 samples) (*Trinkl et al., 2020*). The fatty acids were analysed *via* gas chromatography/mass spectrometry (GCMS; Supplemental Materials).

We quantified the amino acids and sugars of the nectar from three plant species and four treatments (12 samples) as described in *Venjakob, Leonhardt & Klein (2020)* at the University of Freiburg in Freiburg, Germany. The analysis of the nectar amino acids and sugars was carried out chromatographically with an HPLC system (Agilent Technologies 1260 Series; Agilent, Böblingen, Germany; Supplemental Materials).

## Data analysis

Our ultimate goal was to determine whether the treatments in the greenhouse resulted in changes in pollen and nectar quality, and whether these changes corresponded to the changes in pollinator visitation we observed in the field. As such, we aggregated the field visitation data over time to each plant species in each treatment.

First, we tested for differences between treatments among plant species in terms of the concentrations of (a) pollen total amino acids (summed concentrations of all amino acids), (b) pollen total fatty acids (summed concentrations of all fatty acids), (c) number of different pollen fatty acids, (d) pollen production per flower, (e) nectar total amino acids, (f) nectar total sugars, and (g) nectar production per flower. We used generalized linear mixed effect models (GLMMs, R package "lme4") with treatment as a fixed effect and

subsample nested within plant species as the random effect (*Bates et al., 2014*). Note these are not true replicates because we pooled pollen and nectar across individuals of a species within treatments from the greenhouse to have a sufficient quantity to analyse. Instead, these numbers represent variation within and between samples relative to variation within our subsamples. We provide results from among species comparisons in the supplement (Table S2).

Next, we tested for a correlation between the pollen or nectar attributes, or between the pollen and nectar attributes and flower visitation in the subgroups laid out above for data aggregated at the species and treatment level. We used visitation rate (abundance of visitors in a given sample divided by the size of the floral display (inflorescence size*number)) as a normalized measure for comparing visitation among plant species with variable floral displays. When visitation rate increases, it indicates a per floral unit preference (*Russo et al., 2019b*; *Russo et al., 2020*). We also tested the relationship between visitation rate and the ratio of proteins:lipids in the pollen (here pollen amino acids *vs* pollen fatty acids).

We separately evaluated the following groups of flower-visiting insects: (1) pollen-collecting bees (females of non-parasitic species, 1,320 observations), (2) all bees (1,755 observations), (3) bumblebees (1,178 observations), (4) honeybees (386 observations), (5) hoverflies (Syrphidae, 677 observations), and (6) all flower-visiting insects (2,567 observations). We hypothesized pollen-collecting bees would be most sensitive to pollen quality because they are provisioning offspring, and that bumblebees would be sensitive to protein:lipid ratios in the pollen as found in previous studies (*e.g.*, *Vaudo et al., 2016a*; *Russo et al., 2019b*).

Next, we tested whether any of the attributes of pollen or nectar significantly improved the fit of the visitation data in the field, compared to published models of pollinator visitation (*Russo et al., 2020*). These tested whether pollinator visitation was influenced by pollen/nectar quality beyond previously established variables. We used a model selection process, choosing the model with the lowest AICc (function dredge in the package "MuMin" *Barton, 2009*). The site and plant species were treated as random effects, while the floral display and experimental treatment were fixed effects (Table 2 for full model structures). We then tested for interactions between the fixed effects in the model with the lowest AICc, and removed fixed effects that were not significant. We reported the marginal and conditional $R^2$ for all models. Because the field visitation data were zero-inflated, we ran two sets of models. First, we ran a model with a binary presence/absence response variable. Next, we ran a model using only samples where flower-visiting insects were recorded, with insect abundance as the response variable. For models with abundance (rather than visitation rate) as a response variable, floral display was included as a fixed effect and we tested for interactions between floral display and other effects. Interactions were dropped from the final model when they were not statistically significant.

## RESULTS

### Pollen

The plants varied in the amount of pollen produced per flower in the greenhouse (Table S1). For *C. vulgare*, *E. hirsutum*, and *H. radicata*, plants exposed to the combination treatment

Russo et al. (2023), *PeerJ*, DOI 10.7717/peerj.15452

**Table 2  Results of GLMM models, significant contrasts shaded and bolded.** We include the model structure (response and its transformation, fixed effects, random effects, and the contrasts). We also include the number of observations by group (by random effect) for each model. The output from the model includes the effect size, $p$ value (significance), and marginal and conditional $R^2$. Marginal $R^2$ expresses the percent of variation in the response explained by the fixed effects in the model, while conditional $R^2$ expresses the percent of the variation in the response explained by the entire model (fixed and random effects together). The effect size refers to the predicted magnitude of the relationship between the fixed effect and change in the response variable. The $t$ value as reported here is the Wald test statistic. The categorical experimental treatments are Control (C), Fertilizer (F), Herbicide (H), and combination (HF) (Table 1). For all categorical treatment effects, contrasts are in comparison to the control (C); negative effects indicate the treatment is less than the control, while positive effects indicate the treatment is greater than the control. Replicates indicate the number of subsamples collected of pollen and nectar, while Species indicates the plant species in the experiment.

| Response | Transformation | Contrast | Fixed effects | Random effects | Family | Observations | Groups | t | Effect size | p | R²m | R²c |
|---|---|---|---|---|---|---|---|---|---|---|---|---|
| Pollen Total Amino Acids | log | C - F | Treatment | Replicate\|Species | Gaussian | 73 | 6 | −1.00 | −0.05 | 0.32 | 0.004 | 0.93 |
| | | **C - H** | | | | | | **−1.98** | **−0.1** | **0.048** | | |
| | | C - HF | | | | | | −1.41 | −0.07 | 0.16 | | |
| Pollen Total Fatty Acids | no | **C - F** | Treatment | Replicate\|Species | Gaussian | 48 | 6 | **−2.00** | **−1.07** | **0.045** | 0.04 | 0.77 |
| | | C - H | | | | | | 0.45 | 0.24 | 0.65 | | |
| | | C - HF | | | | | | −1.57 | −0.84 | 0.12 | | |
| Pollen per flower | none | **C - F** | Treatment | Species | Gaussian | 24 | 6 | **2.33** | **0.36** | **0.02** | 0.04 | 0.88 |
| | | C - H | | | | | | 0.37 | 0.06 | 0.71 | | |
| | | C - HF | | | | | | 1.79 | 0.27 | 0.07 | | |
| Nectar Sugar | none | C - F | Treatment | Species | Gaussian | 12 | 3 | 1.47 | 13.86 | 0.14 | 0.05 | 0.87 |
| | | C - H | | | | | | 1.91 | 17.98 | 0.06 | | |
| | | C - HF | | | | | | 1.65 | 15.57 | 0.10 | | |
| Nectar Total Amino Acids | log | **C - F** | Treatment | Species | Gaussian | 12 | 3 | **3.94** | **0.40** | **<0.001** | 0.02 | 0.99 |
| | | **C - H** | | | | | | **2.12** | **0.21** | **0.03** | | |
| | | C - HF | | | | | | −0.34 | −0.03 | 0.73 | | |
| Nectar per flower | none | **C - F** | Treatment | Species | Gaussian | 940 | 4 | **2.01** | **0.07** | **0.04** | 0.002 | 0.7 |
| | | C - H | | | | | | 1.07 | 0.04 | 0.28 | | |
| | | **C - HF** | | | | | | **2.10** | **0.07** | **0.04** | | |
| Flower-visitor Abundance | Binary | continuous | Pollen total amino acids | Site, Species | binomial (logit link) | 1,815 | 8, 6 | **3.01** | **0.6** | **0.003** | 0.75 | 0.79 |
| | | | log (Display) | | | | | **9.62** | **3.51** | **<0.001** | | |
| Flower-visitor Abundance | Binary | continuous | Nectar total amino acids | Site, Species | binomial | 944 | 8, 3 | **5.51** | **0.06** | **<0.001** | 0.44 | 0.47 |
| | | | log (Display) | | | | | **14.12** | **1.20** | **<0.001** | | |

Russo et al. (2023), *PeerJ*, DOI 10.7717/peerj.15452
**Table 2** (*continued*)

| Response | Transformation | Contrast | Fixed effects | Random effects | Family | Observations | Groups | t | Effect size | p | R²m | R²c |
|---|---|---|---|---|---|---|---|---|---|---|---|---|
| Flower-visitor Abundance | log presence only | C - F | Treatment | Site, Species | gaussian | 578 | 8, 3 | **2.50** | **0.16** | **0.01** | 0.41 | 0.55 |
| | | C - H | | | | | | **−2.09** | **−0.14** | **0.04** | | |
| | | C - HF | | | | | | −0.27 | −0.02 | 0.78 | | |
| | | continuous | log (Display) | | | | | **17.47** | **0.40** | **<0.001** | | |
| Flower-visitor Abundance | log presence only | continuous | Pollen total amino acids | Site, Species, Sample Date | Gaussian | 876 | 8, 6, 17 | **−4.21** | **−0.51** | **<0.001** | 0.45 | 0.57 |
| | | | log (Display) | | | | | 1.52 | 0.07 | 0.13 | | |
| | | | interaction | | | | | **6.37** | **0.02** | **<0.001** | | |

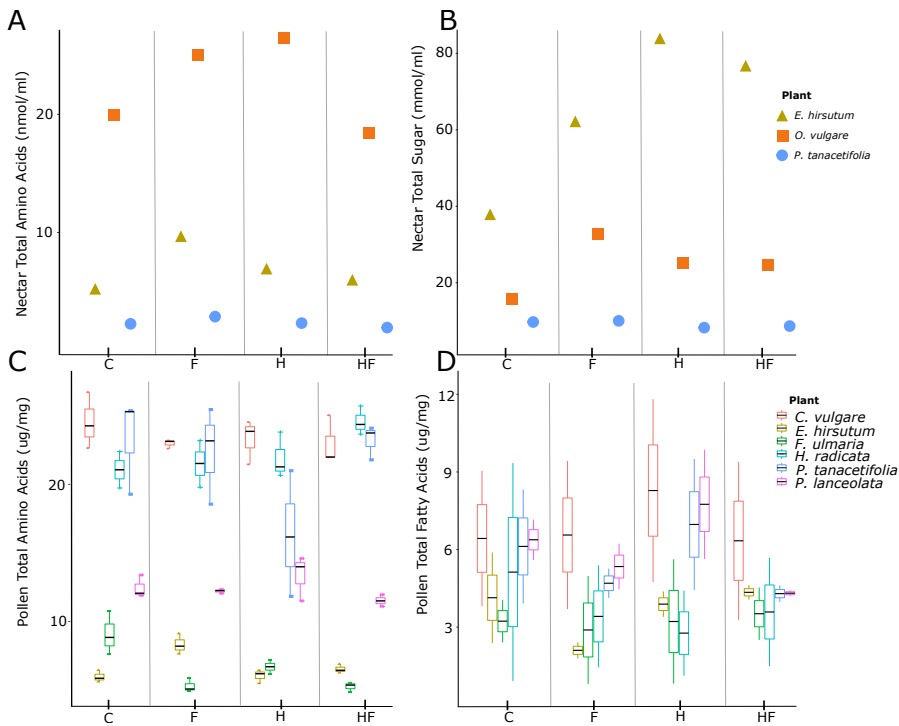

**Figure 1** **Plots of nectar amino acids and sugars and pollen amino acids and fatty acids.** Nectar amino acid (A) and sugar (B) concentrations and pollen amino acid (C) and fatty acid (D) concentrations, separated by plant species (colours) and treatments (C –control, F –fertilizer, H –herbicide, HF –both herbicide and fertilizer). The three species for which we evaluated nectar were *E. hirsutum* (yellow triangle), *O. vulgare* (orange square), and *P. tanacetifolia* (blue circle). The plant species for which we evaluated pollen were *C. vulgare* (red), *E. hirsutum* (yellow), *F. ulmaria* (green), *H. radicata* (light blue), *P. tanacetifolia* (purple), and *P. lanceolata* (pink).

(fertilizer and herbicide) produced the most pollen per flower (Table S1, Fig. S1A). For *F. ulmaria*, *P. tanacetifolia*, and *P. lanceolata*, plants exposed to fertilizer only produced the most pollen per flower. Overall, plants exposed to fertilizer produced significantly more pollen per flower than the control (effect size = 0.36, $t = 2.33$, $p = 0.02$, Table 2).

Herbicide-exposed plants had a significantly lower pollen amino acid concentration than the control (effect size = −0.13, t = −2.52, $p = 0.01$, Fig. 1C). There was a negative effect of fertilizer exposure on the concentration of pollen fatty acids (effect size = −1.07, t = −2.00, $p = 0.045$, Table 2, Fig. 1D). There were significant species level differences in pollen amino acids and fatty acids (Table S2).

## Nectar

Plants exposed to fertilizer or a combination of fertilizer and herbicide produced more nectar per flower than the control, but plants exposed to herbicide alone did not differ from the control (Table 2, Fig. S1B). We found glucose, fructose, and sucrose in the nectar. Concentrations of fructose and glucose correlated with one another and the total sugar concentration, but sucrose concentrations did not correlate with the concentrations of

the other sugars or total sugar concentration. There was no effect of the experimental treatments on total nectar sugar concentrations (Fig. 1B).

Both fertilized plants and those exposed to herbicide had a higher nectar amino acid concentration than the control, while the combination treatment had the same concentration as the control (Table 2, Fig. 1A). There were significant species level differences in nectar sugar and amino acids (Table S2).

## Flower visitation in the field

For the binary model (presence/absence of flower visitors during a sample), the top model included pollen amino acid concentration and log of the floral display as fixed effects, and site and species as random effects. This model did not differ significantly from a model that included treatment as a fixed effect. However, treatment was not significant in this model, so we removed it. This model showed the strongest effect on the presence or absence of a flower-visiting insect during a given sample was the log-transformed floral display (effect size $= -1.14$, $z = 16.94$, $p < 0.001$), but the effect of pollen amino acid concentration was also significant (effect size $= 0.07$, $z = 2.83$, $p = 0.005$, Table 2).

For the binary model, for the plants for which we had nectar data, the top model included nectar amino acids (effect size $= 0.06$, $z = 5.51$, $p < 0.001$) and log of floral display (effect size $= 1.2$, $z = 14.12$, $p < 0.001$, Table 2).

For the presence-only abundance models, we log-transformed abundance. The best model had both pollen amino acids and log of floral display as fixed effects and site and species as random effects. There was a significant interaction between floral display and pollen amino acid concentration (Table 2). At low levels of floral display, pollen amino acids did not affect abundance, but at higher display values, plants with greater pollen amino acid concentration had a greater visitor abundance (Fig. 2). For plants from which we obtained nectar, the best presence-only model had experimental treatment and floral display as fixed effects and did not include any of the nutritional elements (*Russo et al., 2020*).

## Correlations with flower visitation rate

Due to the background effect of floral display (Fig. 2, Fig. S2A), we calculated visitation rate, a display standardized measure of abundance. There was a positive correlation between visitation rate of all flower-visiting insects and nectar amino acids and pollen amino acids and fatty acids, but no relationship between visitation rate and nectar sugar (Fig. 3). The raw abundance data showed a similar positive correlation between pollen amino acid concentration and visitor abundance (Fig. S2E), but negative correlations between abundance and nectar sugar (Fig. S2B) and nectar amino acids (Fig. S2C), and no correlation between abundance and pollen fatty acids (Fig. S2F, Fig. 4).

The visitation rate of all pollinator subsets correlated significantly with the concentration of nectar amino acids, and all but honeybees correlated significantly with the concentration of pollen amino acids (Fig. 5). None of the pollinator subgroups had a visitation rate that correlated with the concentration of nectar sugars, but the abundance of bumblebees, all bees, and pollen-collecting bees all increased with pollen fatty acid concentrations.

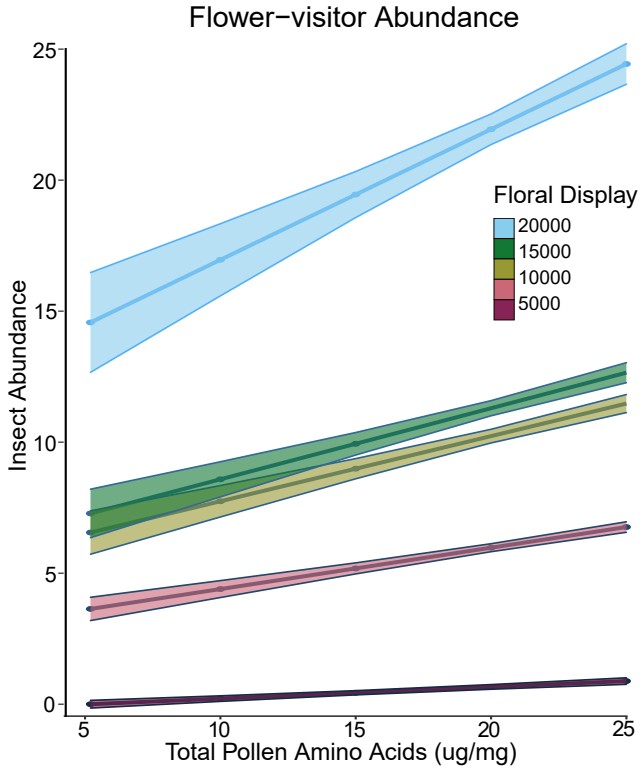

**Figure 2** **Interaction plot of floral display and pollen amino acids *vs* insect abundance.** Interaction plot of the predicted relationship between the total amino acid concentration of the pollen (*x*-axis) and the abundance of flower-visiting insects (*y*-axis). As the size of the floral display increases, the effect of the amino acid concentration in the pollen also increases. Darker colours indicate a smaller floral display, while lighter colours indicate a larger floral display.

The protein:lipid ratio (amino acids/fatty acids) did not correlate with visitation rate in any subgroup except Syrphidae (Fig. 5). However, using GLMMs, the protein:lipid ratio (amino acids/fatty acids) and pollen amino acids were significant predictors of visitation rate across all flower-visitors (Table S3), and protein:lipid ratio, pollen amino acids, and pollen fatty acids were all significant predictors of bumble bee visitation, while protein:lipid ratio and pollen amino acids were predictors of visitation rate of just bees (Table S3).

## DISCUSSION

The results of our study support a relationship between agrochemical (herbicide/fertilizer) exposure, floral resource quality (*i.e.,* pollen and nectar nutritional value), and flower visitors. These findings suggest not only that agrochemical exposure can significantly alter the quality of floral resources, but also that flower-visiting insects are sensitive to these changes. Thus, it appears flower-visiting insects respond to both inter- and intraspecific variation in floral resource quality. In the following, we discuss which nutritional measures in pollen and nectar responded to the studied agrochemicals.

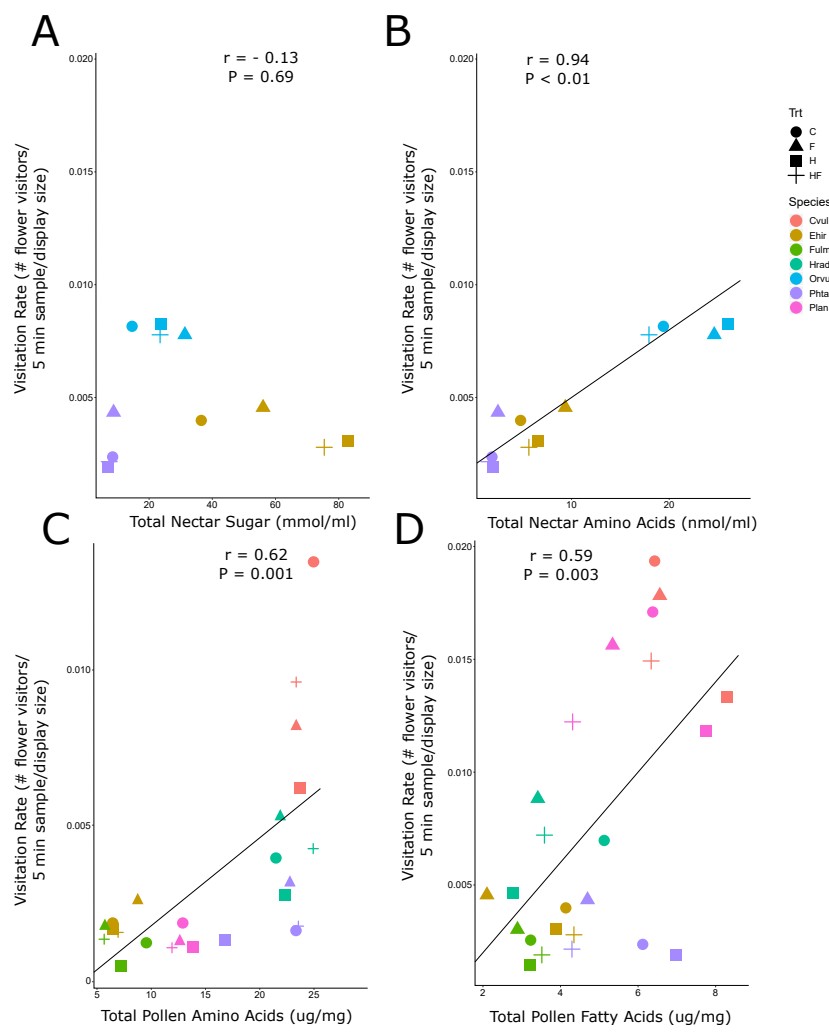

**Figure 3 Correlation plots of nectar sugars and amino acids and pollen amino acids and fatty acids with insect visitation rates.** The correlation between visitation rate (number of flower-visiting insects per unit floral area per five minute sample), and the total nectar sugar (A), total amino acids (B), total pollen amino acids (C), and total pollen fatty acids (D). The Pearson correlation coefficient (r) and significance values (P) are provided for each relationship. The four experimental treatments are Control (C: circle), Fertilizer (F: triangle), Herbicide (H: square), and Combination (HF: cross). The plant species are *C. vulgare* (red), *E. hirsutum* (yellow), *F. ulmaria* (green), *H. radicata* (light blue), *O. vulgare* (blue), *P. tanacetifolia* (purple), *P. lanceolata* (pink).

## Effects of herbicide and fertilizer exposure on nectar and pollen quantity and quality

Although the strongest determinant of pollen and nectar amino acid, fatty acid, and sugar concentrations was species identity, experimental exposure to even very low concentrations of fertilizer and herbicide had significant effects on the nutritional attributes of both nectar and pollen. Herbicide exposure negatively affected the total pollen amino acid concentration, while fertilizer negatively affected the total pollen fatty acid concentration. Both fertilizer and herbicide-exposed plants had higher nectar amino acid concentrations

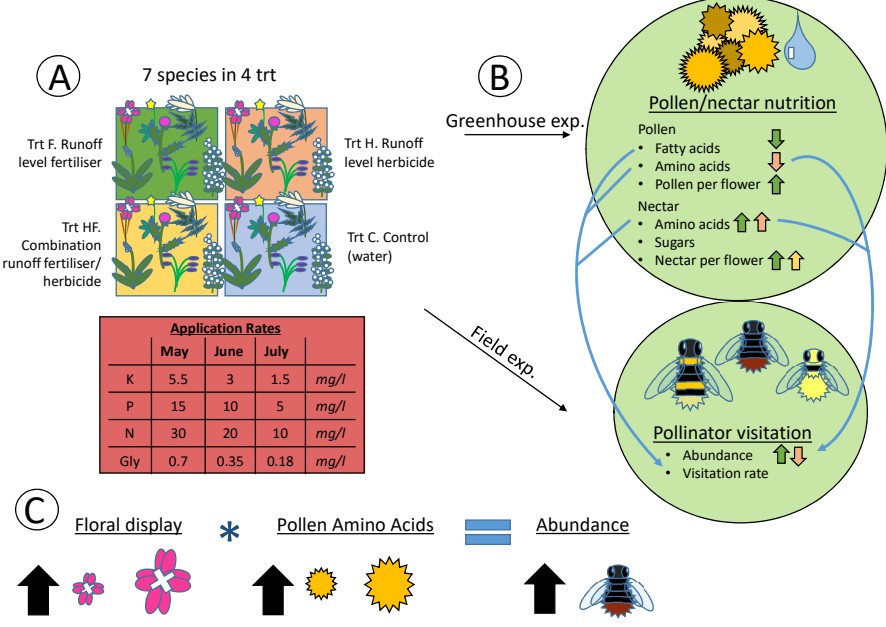

**Figure 4 Heuristic diagram of experimental design and treatment effects.** Flow diagram showing experimental set up and relationships between treatments and pollen and nectar nutrition, along with pollinator visitation. The experiment consisted of seven species in four agrochemical treatments (Trt), the concentrations of which decreased over time (A). These species and treatments were replicated both in the greenhouse and field at the same time, with pollen and nectar attributes being collected in the greenhouse and pollinator visitation being collected in the field (B). Significant effects of the experimental treatments on the pollen and nectar attributes are indicated with arrows (positive ↑, negative ↓) colored by the treatment (green = fertilizer, red = herbicide, yellow = combination). Significant positive correlations between pollen and nectar and pollinator visitation are indicated with blue arrows connecting the greenhouse and field experiments (B). Finally, the model with the best fit for explaining pollinator abundance included the size of the floral display, the pollen amino acid concentration, and an interaction between the two (C). As floral display and pollen amino acid concentrations increase, so does pollinator abundance.

than the control. A combination of herbicide and fertilizer did not affect any of the nutritional attributes of the pollen or nectar. This may be due to the fact that these two agrochemicals had contrasting effects on plant resource allocation and largely balanced one another in this experiment. While the plant species identity had a stronger explanatory effect in all cases, the fact that these low volumes of agrochemical exposure altered fundamental nutritional attributes of floral resources has important implications for pollinator health.

The implications for flower-visiting insects as a result of changes in floral resources quality are unclear, but research suggests significant impacts for pollinator health (*Lau et al., 2022*). Some studies have shown a decrease in fitness and body size of pollinating insects in agricultural systems (*Centrella et al., 2020*). Moreover, pollen nutrient composition has been shown to affect bumblebee colony development (*Moerman et al., 2017*), while solitary bees have been shown to mix unfavorable pollens with favorable pollens to improve nutrition (*Eckhardt et al., 2014*). Similarly, low nectar sugar and low protein have both been shown to constrain larval growth in solitary bees (*Burkle & Irwin, 2009*) and honeybees (*Nicholls, Rossi & Niven, 2021*), respectively. Decreased floral resource quality,

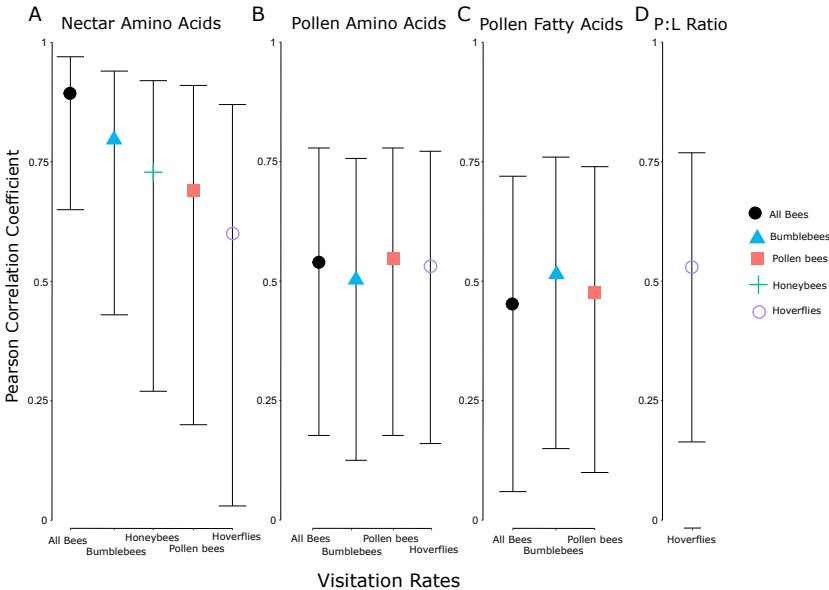

**Figure 5** **Correlation coefficients between nutritional attributes and insect subgroups.** Pearson correlation coefficients for significant relationships between visitation rates of subsets of different groups of flower visitors and nectar amino acids (A), pollen amino acids (B), pollen fatty acids (C), and the ratio between pollen amino acids and fatty acids (protein:lipid ratio) (D). Error bars indicate 95% confidence intervals around the correlation coefficients. The individual insect groups evaluated here include all bees (males and females of Apoidea), bumblebees (males and females of *Bombus* spp.), pollen bees (females of pollen-collecting bee species), honeybees (*Apis mellifera*), and hoverflies (flies in the family Syrphidae).

aside from quantity, could be one mechanism behind observed changes in pollinator health over landscape scales (*Kaluza et al., 2018*; *Parreño et al., 2021*). In addition, exposure to agrochemicals could alter other attributes of the interaction between plants and flower-visiting insects, including electrical signals (*Hunting et al., 2022*), susceptibility to parasites (*Baron, Raine & Brown, 2014*), and decreased foraging efficiency (*Boff et al., 2022*).

A study exploring the effect of drift level exposure to a different herbicide (dicamba and 2,4-dichlorophenoxyacetic acid) found protein levels in pollen were not affected, but the number of flowers produced by the exposed plants was lower than a control (*Bohnenblust et al., 2016*). In our study, we did not observe significant differences in the number of flowers produced in different treatments, and herbicide-exposed plants did not differ from the control in terms of the quantity of pollen or nectar produced per flower (*Schmitz, Schäfer & Brühl, 2013*). However, plants exposed to fertilizer produced larger quantities of both pollen and nectar per flower than non-exposed plants. Because they did not produce significantly more flowers than non-exposed plants, they may have invested in floral reward rather than floral display. Plants exposed to the combination treatment of both fertilizer and herbicide also produced more nectar per flower, but not pollen per flower, than the control.

## Relationship between nectar and pollen composition and flower visitation

When it came to chemical aspects of pollen and nectar and corresponding insect visitation, we observed negative effects of herbicide exposure (*e.g.*, decreases in pollen amino acids and visitation), positive effects of fertilizer (*e.g.*, on pollen and nectar production and visitation), and mostly balanced effects of the combination treatment.

Adding information about both pollen and nectar nutrition improved the model fit for the binary presence/absence models of pollinator visitation. The pollen amino acid concentration correlated with visitation, especially when the floral display was large. This agrees with previous work showing preference might relate to pollen protein concentrations in a field setting across the whole pollinator community (*Russo et al., 2019b*). Visitation to plants with small floral display sizes was very low, so it is possible that these plants were already not preferred. Other studies have shown that pollen protein positively correlates with the dependence of flowering plants on pollinating insects, suggesting that there may be a relationship between pollinator attraction and pollen protein (*Ruedenauer et al., 2019b*). Pollen fatty acids and nectar nutrition (sugar and amino acid concentrations) did not improve the fit of the presence-only abundance models of visitation. It is possible these were not strong drivers of pollinator preference, or variation in the species we studied was not great enough to illustrate any effect. Some work has also shown that some amino acids are deterrents in sucrose solution, while others are attractants (*Simcock, Gray & Wright, 2014*). Moreover, we used very low concentrations of fertilizer and herbicide; one might expect higher concentrations of these agrochemicals to show stronger effects.

When we controlled for floral display size by using visitation rate, there were correlations between the average visitation rate of all flower-visiting insects and nectar amino acid concentration and pollen fatty acid concentration, but not nectar sugar concentration or pollen amino acid concentration. Subgroups of pollinating insects responded differently to pollen and nectar attributes. For example, bees and nectar amino acids showed the strongest relationship, but bees also responded significantly to amino acids and fatty acids in the pollen, whereas hoverflies responded less strongly to amino acids in nectar and pollen, and honeybees only responded to the amino acids in the nectar. Chemosensory research on one species of hoverfly (*Eristalis tenax* L.) showed they responded only to the amino acid proline, and may be unable to taste other amino acids (*Wacht, Lunau & Hansen, 2000*). Similarly, previous research showed honeybees may respond more strongly to amino acids than sugar in nectar, in a laboratory setting (*Bertazzini et al., 2010*). Bumblebees and honeybees may also differ in their ability to taste or detect different amino acids (*Ruedenauer et al., 2019a*). Interestingly, we did not see any significant correlations between nectar sugar and visitation, while even very small concentrations of amino acids in nectar correlated with visitation among several insect groups. Some bees prefer intermediate sugar levels (*Waller, 1972*) or nectar viscosity may play a mechanistic role because bees may be physiologically limited in their ability to collect nectar above a certain viscosity (*Lechantre et al., 2021*). Among our plant species, protein seemed to be more strongly correlated with pollinator preference than sugar, potentially because of the fitness implications of pollen protein levels. As other research supports (*e.g.*, *Lau et al., 2022*), floral resource nutrition is multidimensional and

complex. Finally, only Syrphidae visitation rates correlated with protein:lipid ratios in the pollen, although more comprehensive GLMMs showed significant relationships with protein:lipid ratios and visitation by all bees and just bumblebees. Because bumblebees comprise the largest abundance of all bees, it is possible that the relationship here between protein:lipid ratios and all bees is driven primarily by bumblebees (*Vaudo et al., 2016a*; *Vaudo et al., 2016b*; *Vaudo et al., 2020*).

In order to collect sufficient quantities of nectar and pollen for our nutritional analyses, and to avoid altering pollinator behavior in the field, we collected these resources from plants grown in a greenhouse. The plants were subjected to the same treatment and water regimes as the field plants, but plants grown in the greenhouse can differ physiologically from field plants. Though we were not able to address this variance in our study, future studies could evaluate the extent to which differences in field and greenhouse conditions mediate the effects of the treatments on floral resource quality. Importantly, in spite of this potential source of variance, we still saw significant associations between differences in floral resource quality due to treatment and species identity in the greenhouse and insect visitation to these same species and treatments in the field.

One aspect that remains to be explored in depth is whether individual amino acids or fatty acids play a strong role in determining pollinator preference (*Bertazzini et al., 2010*). There may be variation in the ability of insects to taste different components (*Wacht, Lunau & Hansen, 2000*; *Ruedenauer et al., 2019a*; *Ruedenauer et al., 2020*), and some amino or fatty acids may be more limiting for insects than others. Bees also appear to vary in their preferences of relative nutritional aspects of pollen (*Kriesell, Hilpert & Leonhardt, 2017*). It will be important to test the fitness implications of this variance in floral resource quality for pollinating insects, especially as their preferences do not always increase their fitness (*Hoover et al., 2012*). Our study did not evaluate other components of pollen and nectar, such as bacteria, sterols, plant secondary metabolites, and micro-nutrients, but they have shown to be important drivers of both flower-visiting insect behaviour and fitness (*Palmer-Young et al., 2018*; *Vannette, 2020*; *Sculfort et al., 2021*; *Filipiak et al., 2022*).

## CONCLUSIONS

As concerns about declining populations of beneficial flower-visiting and pollinating insects grow, we must learn to promote healthier pollinator populations through improved nutrition. Understanding the multi-faceted effects of land-use change on these insects includes exploring the importance of agrochemicals on the quality of the resources these insects can access. Multiple studies have shown the importance of floral nutrition in pollinator preference, as well as the importance of floral resource quality for pollinator health (including fitness, immune health, and body size). As we demonstrate here, flower-visiting insects are sensitive to both inter- and intraspecific variation in the quality of these floral resources. Pollinators depend on weedy field-edge habitat in agricultural systems, and the ability of these plants to provide high quality floral resources is affected by stressors, such as agrochemical exposure. Pollinator health will thus likely best be served by high quality habitat protected from chemical stress.

## ACKNOWLEDGEMENTS

We would like to thank J. Stone and S. McNamee for help in the glasshouse project. We are extremely grateful to all of the sites that allowed us to use space on their land to conduct our study, as well as access to their water taps, including University College Dublin and the Lamb Clarke Irish Historical Apple Collection at Rosemount Environmental Research Station, Gas Networks Ireland, Raidi'o Teilifís Éireann, Trinity College Dublin, the Marino Institute, Riverview Educate Together National School, and Airfield Estate for access to sites for field research. We give special thanks to Dr. W. Deasy and B. Moran (UCD Rosemount ERS), Dr. K. McAdoo (Airfield Estate), T. Bannon (RTE), C. van der Kamp and R. Hession (GNI), C. Bennett and E. Kavanagh (UCD Estates and Facilities), C. Fogarty and S. Austin (Marino Institute), M. Burke and R. Judge (Riverview ETNS), S. Waldren, E. Bird, and M. McCann (Dartry Botanic Gardens) for technical assistance, S. Palumbo, A. Flaherty, B. Malone for field help, and O. Fenton, D. OHuallachain, J. Finn, J. Zimmerman, J. Parnell, and S. Hodge for advice. Species identifications of the bees were verified at the National Biodiversity Data Centre in Waterford, Ireland with help from Ú. Fitzpatrick and T. Murray, while syrphid fly identifications were corrected and validated by M. Speight, of Trinity College Dublin. For insects other than syrphid flies and bees, we received additional assistance from specialists M. Smith and B. Nelson (National Parks and Wildlife Service).

### Funding

Funding for this study was provided by a Marie Curie Independent Fellowship (grant number FOMN-705287) to Laura Russo and Jane C. Stout. The funders had no role in study design, data collection and analysis, decision to publish, or preparation of the manuscript.

### Grant Disclosures

The following grant information was disclosed by the authors:
Marie Curie Independent Fellowship: FOMN-705287.

### Competing Interests

The authors declare there are no competing interests.

### Author Contributions

- Laura Russo conceived and designed the experiments, performed the experiments, analyzed the data, prepared figures and/or tables, authored or reviewed drafts of the article, and approved the final draft.
- Fabian Ruedenauer analyzed the data, prepared figures and/or tables, authored or reviewed drafts of the article, and approved the final draft.
- Angela Gronert analyzed the data, prepared figures and/or tables, authored or reviewed drafts of the article, and approved the final draft.

- Isabelle Van de Vreken analyzed the data, authored or reviewed drafts of the article, and approved the final draft.
- Maryse Vanderplanck analyzed the data, prepared figures and/or tables, authored or reviewed drafts of the article, and approved the final draft.
- Denis Michez conceived and designed the experiments, authored or reviewed drafts of the article, provided resources, and approved the final draft.
- Alexandra Klein conceived and designed the experiments, authored or reviewed drafts of the article, provided resources, and approved the final draft.
- Sara Leonhardt conceived and designed the experiments, authored or reviewed drafts of the article, provided resources, and approved the final draft.
- Jane C. Stout conceived and designed the experiments, authored or reviewed drafts of the article, provided resources, and approved the final draft.

## Data Availability

The data are available at DataDryad: Russo, Laura et al. (2023), Fertilizer and herbicide alter nectar and pollen quality with consequences for pollinator floral choices, Dryad, Dataset, https://doi.org/10.5061/dryad.qnk98sfmd.

## Supplemental Information

Supplemental information for this article can be found online at http://dx.doi.org/10.7717/peerj.15452#supplemental-information.

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
