# Peer review of "Fertilizer and herbicide alter nectar and pollen quality with consequences for pollinator floral choices"

_PeerJ, doi:10.7717/peerj.15452_

## Round 0.1 · original submission · Minor Revisions

Dear Dr. Russo,

Congratulations on the first review round of your manuscript! After the assessment of three reviewers, your manuscript received three "minor review" statuses, attesting to the quality of the research in it.

I agree with all the minor reviews that should be made to your study. As soon as you have prepared the new version of the manuscript along with a rebuttal letter to your reviewers, I will be pleased to forward it back to the reviewers, for the second (and, probably, final) review round.

Sincerely,
Daniel Silva

·

Basic reporting

No comment

Experimental design

No comment

Validity of the findings

No comment

Additional comments

The authors present a manuscript for a study to determine the upstream effects of fertilizer and herbicide applications on the resources plants produce for pollinators. This study builds on the existing literature on pollinator nutrition and incorporates relatively novel effects on how the resources plants produce can be manipulated, which are understudied. The manuscript is well written and logically organized. The experimental design and statistics used are also sound. I only have a few comments and suggestions that the authors may want to consider to further improve their manuscript.


Lines 134-135: How much nectar was needed for nutritional analyses? This was also not described in the supplemental methods

Lines 137-138: Plants grown in the greenhouse can vary physiologically quite substantially compared to natural conditions in the field despite mimicking as many environmental conditions as possible and there are a few studies that show that greenhouse vs field data is not correlated. I understand that a comparison would be outside the scope of this study, but this is just something I wanted to point out as a note for future studies or something to note in this one.

Line 141: Describe pollen collection techniques in more detail. For the plants where the entire anthers were collected, were they included in the nutrient analyses? If so, are there concerns that the plant material is diluting the nutrient content detected in pollen? How were they collected? Forceps? Vacuum?

Line 283: Overall fatty acids are certainly important. However, I would like to see the authors expand on this a bit more considering their previous research on the importance of different fatty acids. I know this would add more complexity to the results, but did you consider looking into the different fatty acid distributions, notably linoleic and linolenic acids?

Line 370: I am surprised to see that there were not significant correlations between nectar sugar and visitations. A significant amount of literature use nectar sugar concentration as a predictor for assessing plant pollinator attractiveness. As this study and the research group's previous work suggest, pollinator nutrition is multidimensional and incredibly complex to understand. I think it would be worthwhile to expand on why this might be in the discussion.

Reviewer 2 ·

Basic reporting

This is an interesting, well-designed and well written study experimentally examining the impact of agrochemical use on floral rewards and pollinator foraging preferences. Overall I thought it was well reported, there are just a few areas, outlined below, where the authors should provide more detail regarding the experimental and analytical methods.
Line 52: it may help the reader to include a few brief examples of ‘industrial practices’ which negatively affect pollinators
Line 72; 78, 83: It would help to specify here or earlier that you are (I presume) using ‘quality’ as a short-hand for nutritional composition of floral resources
Data analysis is described in excellent detail.
Figure 1: Something seems to have happened here with the formatting of the y-axes, please correct. Please also indicate the sample size in the figure legend.
Fig 2- this is a neat way to show the interaction between floral display and pollen amino acids on insect abundance, however it is a little difficult to discriminate between the colours in the legend, might using different colours rather than a gradient of colours help?
Figure 1 and Fig. S1- please clarify if this is the mean, median etc and indicate what the box limits, whiskers, error bars etc. denote
Given the complexity of the design and factors measured, the paper might benefit from a stylised diagram/flow chart which summarises the findings of 1) the impact of the different treatments on pollen and nectar characteristics and 2) the subsequent impacts on pollinator visitation

Experimental design

Really great justification for the treatment concentrations and simulation of how this changes throughout the growing season.
Line 144: Did you bag flowers before collecting nectar to try to standardise volume? Were insects able to access the flowers in the greenhouse and if so, might that have affected nectar volumes? Nectar volume/concentration is also known to vary according to temperature and humidity, you reference some papers that have demonstrated this- did the greenhouse set up allow you to control for these environmental conditions? Presumably you sampled the same plants from different treatments on the same days?
Line 139; 181: More detail is needed here on the pollen and nectar sampling for nutritional analysis. How many plants per species per treatment were sampled? Were multiple flowers on the same plant sampled? What is meant by a sub-sample? Is this pollen from three different plants or three different inflorescences on the same plant? Line 205 would suggest you pooled across plants but this should also be outlined in the methods.
Line 180: Given the variation in rewards that may exist between individual plants of the same species (Line 72) what was the justification for the relatively low number of replicates for the pollen (2-3 samples per plant species per treatment) and nectar analysis (1 sample per plant species per treatment)?
Line 204: Again, what is meant by a subsample?

Validity of the findings

My main concern regarding the validity of the findings relates to my comment regarding the relatively small number of samples used in the nutritional analyses. Since only one nectar sample per species, per treatment was analysed, more caution should be used when interpreting the impact of the treatments on reward nutritional quality and nectar volume. See also my comments regarding standardising environmental conditions when sampling nectar.
You should also make it clearer in your discussion that you weren’t directly measuring the pollen/nectar from the plants that you were studying visitation in. These were grown outside and therefore may potentially have responded differently to the treatments and this caveat should be made clear.
Line 318: The fact that the combination of herbicide and fertiliser had no effect is an interesting result that deserves attention- why do you suspect this might be?
In general, the discussion is rather brief and would benefit from more links to existing research and greater consideration of the implications of the findings, for example:
Line 324: ‘results are unclear’- here you could draw on data from nutritional studies that have shown the effect of suboptimal nutrition on both bee colony growth and reproduction (e.g. Moerman et al. 2017 Cons. & Diversity, Eckhardt et al. 2014 J. Animal Ecol., Burkle & Irwin 2009 Env. Entomol.) and individual physiology (e.g. Nicholls et al. 2022 JEB). Similarly in Line 381.
In your discussion of floral rewards and flower visitation, you could draw more links between your findings and previous behavioural evidence e.g. how do bigger rewards typically impact on pollinator floral choices.
371: Nectar viscosity may play a mechanistic role- can you expand on what you mean here
Some references that are missing discussion in the discussion section include Vaudo et al. 2016 and Vaudo et al. 2020

Reviewer 3 ·

Basic reporting

no comment

Experimental design

no comment

Validity of the findings

no comment

Additional comments

I must say that I have enjoyed reading your new manuscript and learned from it how environmental stressors can impact plants traits.

Below I present further considerations in your manuscript.
Line 33: “and collected pollen and nectar from selected plants in the …”
Line 94-100. Can plants be included in different groups according to the resources they offer or they all offer pollen and nectar as resource to pollinators?
Line 162. Why did you consider individuals that have contacted reproductive parts? Did you not consider those visitors that visited the plant and have explored resources without touching the reproductive parts (if any)?
Line 188. Were fatty acids separated in different categories (saturated and unsaturated)? Any major effect of treatment in these two categories that should be considered important for pollinators?
Line 213. What is meant by inflorescence size? Did you measured it in cm from the first flower to the most distant one in the inflorescence? please clarify.
Line 226. To clarify. Is here also visitation rate (visits divided by size of floral display) presence/absence or total number of visits? I suppose here you did not control for floral display.
Line 280. I think this sentence is a bit loose in the text. Maybe you can include this sentence immediately before showing the related results.
Figure 4. Perhaps I have missed, what do you mean with "pollen bees"?
Line 327. “aside from quantity”, see suggested references below.
Line 373. Is there any explanation for protein be more correlated than sugar in your study?
Line 375. what happen if you remove bumblebees from the analysis? I wonder if the interaction is actually driven only by bumblebees (1178 observations).
Literature suggested:
Ellard R Hunting, Sam J England, Kuang Koh, Dave A Lawson, Nadja R Brun, Daniel Robert, Synthetic fertilizers alter floral biophysical cues and bumblebee foraging behavior, PNAS Nexus, Volume 1, Issue 5, November 2022, pgac230, https://doi.org/10.1093/pnasnexus/pgac230
Baron GL, Raine NE, Brown MJ. Impact of chronic exposure to a pyrethroid pesticide on bumblebees and interactions with a trypanosome parasite. Journal of Applied Ecology. 2014 Apr;51(2):460-9.
Boff S, Keller A, Raizer J, Lupi D. Bumble bee workers face decreased efficiency of pollen collection and reduction in size due to Sulfoxaflor exposure in late European summer. Frontiers in Ecology and Evolution.:800.

---

## Round 0.2 · Minor Revisions

Dear Dr. Russo,

Although this version of your manuscript has just received a "minor review" status, in practice, after you take care of the small issues raised by both reviewers, your manuscript will be formally accepted for publication in PeerJ. Congratulations on your hard work!

Sincerely,
Daniel Silva

·

Basic reporting

The authors addressed all the minor comments and suggestions made by the reviewers, and the manuscript is suitable for publication. The new figure is also a great addition. I only have one thing to add to reviewer 3's comment on the explanation for protein being more correlated to sugar (line 373). I suggest looking into Dr. Geraldine Wright's work on this. Their group has done work on bee amino acid balancing in nectar. There might be some papers you can cite from there as a possible explanation, and that might be more appropriate the the Vaudo papers since they looked at P:L. Besides that, congrats to the authors!

Experimental design

N/A

Validity of the findings

N/A

Reviewer 3 ·

Basic reporting

no comment

Experimental design

no comment

Validity of the findings

no comment

Additional comments

A doubt that emerge just now relates to the quantity of pollen increasing with fertilizer. My question is: Do authors controlled for flower sizes? Probably they are correlated measures but I think it would be interesting to know if the plants produce more pollen (and nectar) because flowers became also bigger or they produced more pollen and nectar independent of flower size. Please clarify.

Please certify to correct your reference list. Some of your cited papers are not in the reference list.

---

## Round 0.3 · accepted · Accept

Dear Dr. Russo,

All issues have been considered in the new version of your manuscript. Therefore, I congratulate you and your co-authors for your hard work and for improving this vital contribution to the entomological research field.

Sincerely,
Daniel Silva, Ph.D.